# Label super-resolution networks

**Kolya Malkin**[1,2]    **Caleb Robinson**[1,3]    **Le Hou**[1,4]    **Rachel Soobitsky**[5]    **Jacob Czawlytko**[5]
**Dimitris Samaras**[4]    **Joel Saltz**[4]    **Lucas Joppa**[1]    **Nebojsa Jojic**[1]*
[1]Microsoft Research    [2]Yale University    [3]Georgia Institute of Technology
[4]Stony Brook University    [5]Chesapeake Conservancy

## Abstract

We present a deep learning-based method for super-resolving coarse (low-resolution) labels assigned to groups of image pixels into pixel-level (high-resolution) labels, given the joint distribution between those low- and high-resolution labels. This method involves a novel loss function that minimizes the distance between a distribution determined by a set of model outputs and the corresponding distribution given by low-resolution labels over the same set of outputs. This setup does not require that the high-resolution classes match the low-resolution classes and can be used in high-resolution semantic segmentation tasks where high-resolution labeled data is not available. Furthermore, our proposed method is able to utilize both data with low-resolution labels and any available high-resolution labels, which we show improves performance compared to a network trained only with the same amount of high-resolution data. We test our proposed algorithm in a challenging land cover mapping task to super-resolve labels at a 30m resolution to a separate set of labels at a 1m resolution. We compare our algorithm with models that are trained on high-resolution data and show that 1) we can achieve similar performance using only low-resolution data; and 2) we can achieve better performance when we incorporate a small amount of high-resolution data in our training. We also test our approach on a medical imaging problem, resolving low-resolution probability maps into high-resolution segmentation of lymphocytes with accuracy equal to that of fully supervised models.

## 1 Introduction

Semantic image segmentation is the task of labeling each pixel in an input image $X = \{x_{ij}\}$ as belonging to one of $L$ fine-scale *application* classes, $Y = \{y_{ij}\}, y \in \{1, \ldots, L\}$. In *weakly supervised segmentation*, instances in the training set only contain partial observations of the target ground truth labels, *e.g.*, summary of class labels instead of pixel-level labels. We aim to solve a variant of this problem where coarse-scale, low-resolution *accessory* classes, $Z = \{z_k\}; z \in \{1, \ldots, N\}$, are defined for sets of pixels in the input images, where we are given the joint distribution $P(Y, Z)$ between the accessory class labels and the *application* labels. Specifically, a training image $X$ is divided into $K$ sets $B_k$, each with an accessory class label $z_k$, and our models are trained to produce the high-resolution *application* labels $y_{ij}$. For example, in Figure 1, a high-resolution aerial image is shown alongside the low-resolution *ground truth* land cover map (defined over accessory classes) and the target high-resolution version (defined over *application* classes). We aim to derive the high-resolution land cover map based on the aerial image and low-resolution ground truth.

Compared to other weakly supervised image segmentation techniques, the formulation of the problem we aim to solve is more general: it applies both to existing weakly supervised image segmentation problems, as well as to other problems with different characteristics of weak labels. The more general formulation is necessary for tasks such as land cover mapping from aerial imagery and lymphocyte segmentation from pathology imagery. In these applications, coarse labels do not necessarily match the fine-scale labels, as shown in Figure 1. The distinction between the fine-scale *application* and coarse-scale accessory classes is necessary for situations in which the ground-truth information that is known about an image does not match with the *application* classes that we aim to

---

*jojic@microsoft.com

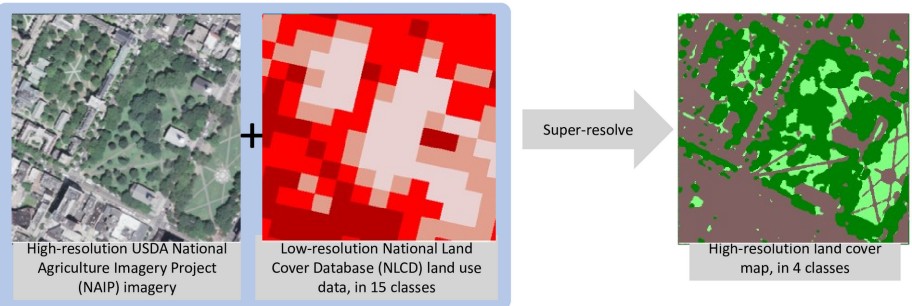

Figure 1: An Illustration of land cover data and label super-resolution. Our method takes an input image ($x$) with low-resolution labels ($z$) and outputs a set of super-resolved label predictions ($y$), utilizing the statistical descriptions between low-resolution and high-resolution labels (Appendix B) *e.g.*, one low-resolution class designates areas of low-intensity development, with 20% to 49% of impervious surfaces (such as houses or roads).

label the image with, but instead suggests a distribution over the application labels. State-of-the-art methods for weakly supervised semantic segmentation exploit the structure of weak labels in ways that are not applicable in our examples: we cannot create bounding boxes around land cover object instances (Dai et al. (2015); Papandreou et al. (2015)) – we consider data that is generally given at scales much larger than the objects being segmented and does not carry foreground-background morphology – nor use coarse approximations of ground-truth segmentation (Krähenbühl & Koltun (2011); Hong et al. (2015)). Other work attempts to match a class "density function" to weak labels (Lempitsky & Zisserman (2010)), but it mainly targets localization and enumeration of small foreground objects with known sizes. Existing Weak supervision approaches also often involve expensive steps in inference, such as CRFs or iterative evaluation (Chen et al. (2015)), which are impractical on large datasets. At the same time, thorough analyses of training algorithms only exist for models that are not sufficiently expressive for the applications we consider (Yu et al. (2013)). While our formulation of the problem allows us to specifically address the previously mentioned land cover mapping and lymphocyte segmentation, it can also be applied to more traditional segmentation tasks such as foreground/background segmentation as we explore in Appendix. F.

Our proposed method is illustrated in Figure 2. Briefly, a standard segmentation network will output probabilistic estimates of the application labels. Our methodology summarizes these estimates over the sets $B_k$, which results in an estimated distribution of application labels for each set. These distributions can then be compared to the expected distribution from the accessory (low-resolution) labels using standard distribution distance metrics. This extension is fully differentiable and can thus be used to train image segmentation neural networks end-to-end from pairs of images and coarse labels.

Land cover mapping from aerial imagery is an important application in need of such methodology. Land cover maps are essential in many sustainability-related applications such as conservation planning, monitoring habitat loss, and informing land management. In Section 3.1 we describe land cover mapping in detail and show how our method creates high-resolution land cover maps solely from high-resolution imagery low-resolution labels, at an accuracy similar to that of models trained on high-resolution labels. We further show how to train models with a combination of low- and high-resolution labels that outperform the high-res models in transfer learning tasks. As low-resolution labels are much easier to collect, and indeed exist over a much wider geographic area in our land cover mapping application, the ability to combine low- and high-resolution labels is an important feature of our proposed methods.

In a second example (Section 3.2), we segment tumor infiltrating lymphocytes from high-resolution (gigapixel) pathology images. Understanding the spatial distribution of immune cells, such as lymphocytes in pathology images, is fundamental for immunology and the treatment of cancer (Finn (2008); Thorsson et al. (2018)). Here, coarse labels are probabilities of lymphocyte infiltration (having two or more lymphocytes) on $100 \times 100$ pixel regions, given by an automatic classifier (Saltz

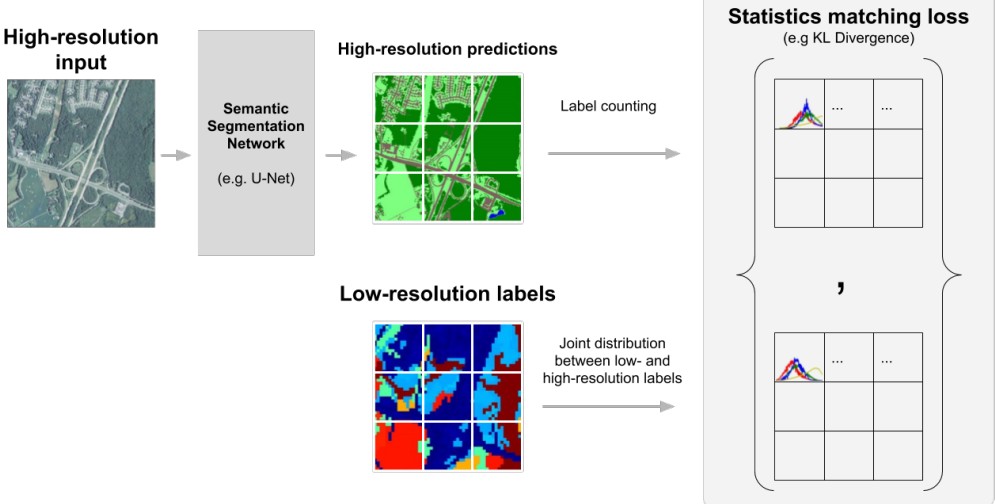

Figure 2: Proposed statistical matching loss function for label super-resolution shown with example images from our land cover labeling application. The model's high-resolution predictions in each low-resolution block are summarized by a label counting layer and matched with the distributions dictated by the low-resolution labels.

et al. (2018)). Our super-resolution model trained on coarse labels performs the same as a lymphocyte classifier trained with high-resolution (cell-level) supervision (Hou et al. (2018)).

To summarize, as our **first contribution**, we propose a label super-resolution network which utilizes the distribution of high-resolution labels suggested by given low-resolution labels, based on visual cues in the input images, to derive high-resolution label predictions consistent to the input image. Our **second contribution** is that we evaluate our method extensively on the application of land cover segmentation and conclude that when there are not enough representative high-resolution training data, our method is much more robust than a model trained on high-resolution training data only, since our method utilizes more training data with weak labels. We show the generality of our method on the lymphocyte segmentation task and the task of segmenting foreground given object bounding boxes (in Appendix F).

## 2 CONVERTING A SEMANTIC SEGMENTATION NETWORK INTO A LABEL SUPER-RESOLUTION NETWORK

A semantic segmentation network takes pixels $X = \{x_{ij}\}$ as input and produces a distribution over labels $Y = \{y_{ij}\}$ as output. If $\phi$ are learned network parameters, this distribution is factorized as:

$$p(Y|X; \phi) = \prod_{i,j} p(y_{ij}|X; \phi), \tag{1}$$

Each $p(y_{ij}|X; \phi)$ is a distribution over the possible labels, $y \in \{1, \ldots, L\}$. Typically, a network would be trained on pairs of *observed* training images and label images, $(X^t, Y^t)$, to maximize:

$$\hat{\phi} = \arg\max_\phi \log \prod_t p(Y^t|X^t; \phi) = \arg\max_\phi \sum_t \sum_{i,j} \log p(y_{ij}^t|X^t; \phi). \tag{2}$$

In this paper, we assume that we do not have pixel-level supervision, $Y^t$, but only coarse accessory (low-resolution) labels $z_k \in \{1, \ldots, N\}$ given on sets (blocks) of input pixels, $B_k$. We also assume a statistical joint distribution over the number of pixels $c_\ell$ of each *application* label $\ell \in \{1, \ldots, L\}$ occurring in a block labeled with an accessory (low-resolution) label $z$, $p_{\text{coarse}}(c_1, c_2, \ldots, c_L|z)$. Our extension of semantic segmentation networks is described in the following three sections.

**Using coarse labels as statistical descriptors.** In computer vision applications, pixel-level labeled data is typically expensive to produce, as is the case of high resolution land cover mapping where high-resolution labels only exist for limited geographic regions. On the other hand, coarse low-resolution labels are often easy to acquire and are readily available for larger quantities of data. Coarse labels can provide weak supervision by dividing blocks of pixels into categories that are statistically different from each other. To exploit this we must formally represent the distribution of high-resolution pixel counds in these blocks, $p_{\text{coarse}}(c|z)$.

For example, in the case of land cover mapping with four types of high-resolution land cover classes[1], the descriptions of labels from the National Land Cover Database (NLCD) – at 30 times lower resolution than available aerial imagery (Homer et al. (2015)) – suggest distributions over the high-resolution labels. For instance, the "Developed, Medium Intensity" class – see Table 3 in the appendix – is described as "Areas with a mixture of constructed materials and vegetation. Impervious surfaces account for 50% to 79% of the total cover". While such a designation does not tell us the precise composition or arrangement of high-resolution labels within a particular "Developed, Medium Intensity" label , it does describe a distribution. One mathematical interpretation of this particular example is

$$c_{\text{imperv}} \sim \text{unif}(0.5, 0.8), \quad c_{\text{forest}} + c_{\text{field}} = 1 - c_{\text{imperv}}, \quad c_{\text{water}} \approx 0.$$

In practice these descriptions should be interpreted in a softer manner (*e.g.*, with Gaussian distributions) that can account for variance in real-world instances of the coarse classes[2].

**Label counting.** Assume $p_{\text{coarse}}(c|z)$, a connection between the coarse and fine labels, has been represented. Suppose we have a model that outputs distributions over high-resolution labels, $p(Y|X)$ given inputs $X$. We must summarize the model's output over the low-resolution blocks $B_k$. Namely, a *label counting layer* computes a statistical representation $\theta_k$ of the label counts in each block $B_k$.

If we sampled the model's predictions $y_{ij}$ at each pixel, the count of predicted labels of class $\ell$ in block $B_k$ would be

$$c_\ell = \frac{1}{|B_k|} \sum_{(i,j) \in B_k} \delta(y_{ij} = \ell). \tag{3}$$

By averaging many random variables, these counts $c_\ell$ will follow an approximately Gaussian distribution,

$$p_{\text{net}}(c_{\ell,k} = c|X) = \mathcal{N}(c; \mu_{\ell,k}, \sigma_{\ell,k}^2),$$

where

$$\mu_{\ell,k} = \frac{1}{|B_k|} \sum_{(i,j) \in B_j} p(y_{ij} = \ell|X, \phi), \quad \sigma_{\ell,k}^2 = \frac{1}{|B_k|^2} \sum_{(i,j) \in B_k} p(y_{ij} = \ell|X, \phi)(1 - p(y_{ij} = \ell|X, \phi)).$$
$$\tag{4}$$

These two parameters for each label $\ell$ constitute the output of each block's label counting layer $\theta_k = \{\mu_{\ell,k}, \sigma_{\ell,k}^2\}_{\ell=1}^L$. Note that treating each count $c_\ell$ as an independent Gaussian variable (given the input $X$) ignores the constraint $\sum_\ell c_\ell = 1$, and more exact choices exist for modeling joint distributions $p_{net}(\{c_\ell\}|X)$; however, we do have $\sum \mu_\ell = 1$ and thus $\mathbb{E}\left[\sum_\ell c_\ell\right] = 1$. In practice, this approximation works well.

**Statistics matching loss.** The coarse labels $z$ provide statistical descriptions for each block $p_{\text{coarse}}(\{c_\ell\}|z)$, while the label counting modules produce distributions over what the segmentation network sees in the block given the high-res input image $X$, $p_{\text{net}}(\{c_\ell\}|X)$. The statistics matching module computes the amount of mismatch between these two distributions, $D(p_{\text{net}}, p_{\text{coarse}})$, which we then use as an optimization criterion for the core segmentation model. Namely, we set

$$C_\ell = \arg\max_{c_\ell} \left[ \sum \log p_{\text{net}}(c_\ell|X) \log p_{\text{coarse}}(c_\ell|z) \right]$$

---

[1] Water, forest, field, and impervious surfaces.

[2] For example, the label "water" is in fact found in the "Developed (medium intensity)" in around 1% of such pixels in Maryland, and the frequency of impervious surfaces may not lie between 50% and 79% due to occasional misclassification and misalignment in the dataset.

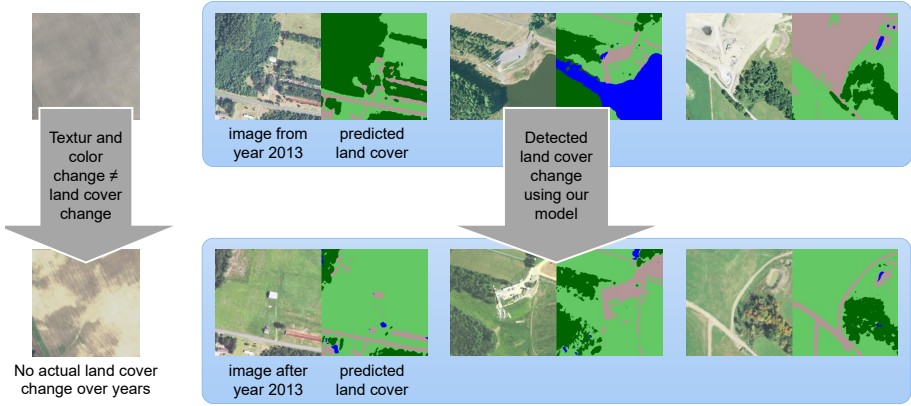

Figure 3: Our model is useful detecting land cover change over years, at the same geographical location, which cannot be achieved effectively by directly comparing satellite images. For a detailed description of how we detect land cover change, see Appendix D.

and seek to maximize $D(p_{\text{net}}, p_{\text{coarse}}) = \sum_l \log p_{\text{net}}(C_\ell | X)$, that is, the likelihood of the distribution of labels $C$ that represents the optimal compromise between what the segmentation network expects (given the image $X$) and what the joint distribution dictates (given the coarse label $z$). In particular, if the distributions $p_{\text{coarse}}(c_\ell | z)$ are also represented as products of Gaussians, *i.e.*,

$$p_{\text{coarse}}(\{c_\ell\}|z) = \prod_\ell \mathcal{N}(c_\ell; \eta_{\ell,z}, \rho_{\ell,z}^2), \tag{5}$$

then

$$C_\ell = \frac{\rho_{\ell,z}^2 \mu_\ell + \sigma_\ell^2 \eta_{\ell,z}}{\sigma_\ell^2 + \rho_{\ell,z}^2}, \tag{6}$$

$$D(p_{\text{net}}, p_{\text{coarse}}) = \log p_{\text{net}}(C_\ell | X) \sim \text{const} - \frac{1}{2} \frac{(\mu_\ell - \eta_{\ell,z})^2}{\sigma_\ell^2 + \rho_{\ell,z}^2} - \frac{1}{2} \log 2\pi \sigma_\ell^2, \tag{7}$$

a function that is differentiable in the output of the label counting layer $\theta = \{\mu_\ell, \sigma_\ell^2\}$. In turn, these are differentiable functions of the input image $X$. Thus, the network can be trained to minimize the sum of the expressions (7) over all blocks $k$ in the input image[3].

## 3 APPLICATIONS AND EXPERIMENTS

### 3.1 LAND COVER SUPER-RESOLUTION

We use our proposed methods in the land cover classification task. Land cover mapping is typically a part automatic, part manual process through which governmental agencies and private companies segment aerial or satellite imagery into different land cover classes (Demir et al. (2018); Kuo et al. (2018); Davydow et al. (2018); Tian et al. (2018)). Land cover data is useful in many settings: government agencies - local, state and federal - use this data to inform programs ranging from land stewardship and environment protection to city planning and disaster response, however this data is difficult and expensive to acquire at the high-resolutions where it is most useful. The Chesapeake Conservancy, for example, spent 10 months and \$1.3 million to generate the first large

---

[3] We could alternatively use the KL divergence, $D(p_{net}, p_{target}) = D_{\text{KL}}(p_{net} \| p_{target})$, as a measure of matching. Then

$$D(p_{\text{net}}, p_{\text{coarse}}) = \text{KL}(p_{\text{net}} \| p_{\text{coarse}}) = \text{const} - \frac{1}{2} \frac{(\mu_\ell - \eta_{\ell,z})^2}{\rho_{\ell,z}^2} - \frac{1}{2} \log 2\pi \sigma_\ell^2.$$

In practice, for large block size, $\sigma_\ell$ is small in comparison to $\rho_{\ell,z}$ (cf. (4)), so our criterion is close to the KL distance, especially late in training.

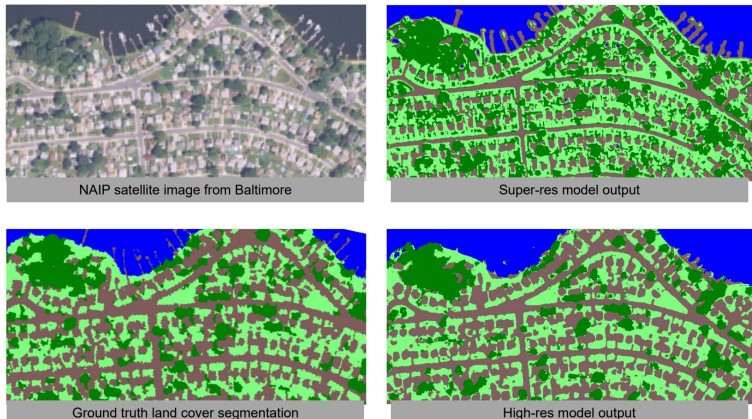

Figure 4: Land cover segmentation examples. The SR model, while never shown pixel-level data in training, finds sharper edges of buildings than the high-res model and even identifies some features along the shoreline that the high-res model misses. For more qualitative examples, see Appendix E.

high-resolution (1m) land cover map for over 160,000 km$^2$ of land in the Chesapeake Bay area (Chesapeake Bay Conservancy (2016; 2017)). Deep learning models that can automate the creation of land cover maps have a large *practical* value. As an example application, we create a method for automated land cover change detection using our models, described in Appendix D (with results in Figure 3). Furthermore, we create an interactive web application that lets users query our best performing models and "paint" land cover maps throughout the US, described in Appendix E.

**Datasets and training.** To demonstrate the effectiveness of our label super-resolution method we have three **goals**: (1) show how models trained solely with low-resolution data and label super-resolution compare to segmentation models that have access to enough representative high-resolution training data; (2) show how models trained using label super-resolution are able to identify details in heterogeneous land cover settings (*i.e.*, in urban areas) more effectively than baseline weakly-supervised models; and (3) show how models trained using a combination of low- and high-resolution data, using our method, are able to generalize more effectively than models which rely on low- or high-resolution labels alone.

We use three datasets: 4-channel high-resolution (1m) aerial imagery from the US Department of Agriculture, expensive high-resolution (1m) land cover data covering the Chesapeake Bay watershed in the north eastern United States (Chesapeake Bay Conservancy (2016; 2017)), and much more widely available low-resolution (30m) NLCD land cover data (see Fig. 1 for examples of the data, and Appendix B). We divide these datasets into four geographical regions: **Maryland 2013 training region** with high-resolution training labels, **Maryland 2013 test region**, **Chesapeake 2013 test region**, and **Chesapeake 2014 test region**. We make the distinction between years the data was collected as weather, time of day, time of year, and photography conditions greatly change the quality of the imagery from year to year – see Fig. 3.

With these datasets we train and test four groups of models: **HR** models which will only have access to high-resolution data in the Maryland 2013 training region, **SR** models, trained with our label-super resolution technique, that only have access to low-resolution labels from the region in which they are tested, **baseline weakly-supervised** models, described in the next section, which will also only have access to low-resolution labels from region in which they are tested, and **HR + SR** models which will have access to the high-resolution labels from Maryland 2013, and low-resolution labels from the region in which they are tested. Given this setup, our experiments will vary two factors:

- *The dataset on which low-resolution data is used, and on which the model is tested.* As low-resolution labeled data is commonly available, we can train models with high-resolution data from the region in which we have it, as well as with low-resolution data from the area that we want our model to generalize to but where high-resolution data isn't available. We simulate this scenario with

high-res data from Maryland and low-res data from the entire Chesapeake, even though the high-res labels are available – and used for our accuracy evaluation – in the rest of Chesapeake as well. This addresses our first two **goals**.

- *The amount of high-resolution data seen in training.* In practical settings it is often the case that a small amount of high-resolution labels exist, as is the case with the land cover data from the Chesapeake Conservancy. To test how well our models will perform under this relaxation, we vary the amount of high-resolution data available from the Maryland 2013 training set from none (*i.e.*, only trained with low-resolution data using our SR models) to all data in the training set. Here, if both low-res and high-res data are used, we jointly optimize the core model on high-res data (using pixelwise cross-entropy loss) and on the low-res data (using our super-res criterion), using a weighted linear combination of the two losses. This addresses our third **goal**.

We use a U-Net architecture as our core segmentation model, and derive the parameters of the joint distributions between accessory (low-resolution) and application labels $(\eta_{\ell,z}, \rho_{\ell,z}^2)$ used in super-res model training from (low-res, high-res) pairs from the Maryland 2013 training set as the true means and variances of the frequencies of high-res label $\ell$ in blocks of low-res class $z$. See Appendices A and B for details on the model architecture/training and joint distribution parameters between low-resolution and high-resolution classes.

**Baseline models.** Our main high resolution baseline model is the U-Net core trained to minimize pixelwise cross-entropy loss using the respective high-resolution labels. U-Net was chosen after experimentation with other standard neural segmentation models: SegNet (Badrinarayanan et al. (2017)), ResNet, and full-resolution ResNet (Pohlen et al. (2017)), all of which achieved overall accuracies from 80 to 83%.

In addition to the high-resolution model, we consider three baseline approaches to weakly supervised segmentation, which we compare to our SR models:

- "Soft naïve": naïvely assigning the NLCD mean frequencies $\eta_{\ell,c}$ as target labels for every pixel in a low-res block and training the core using cross-entropy loss as above.

- "Hard naïve": Doing the same, but using a one-hot vector corresponding to the most frequent label in a given NLCD class ($\arg\max_{\ell} \eta_{\ell,z}$) as the target.

- An EM approach as in Papandreou et al. (2015): (1) M-step: train the super-res model only; (2) E-step: perform inference of high-res labels on the training set, followed by superpixel denoising (average predictions in each block); finally, assign labels in each block according to this smoothed prediction; (3) Repeat the EM iteration. Note that we use superpixel denoising instead of dense-CRF proposed by Papandreou et al. (2015), due to large computational overhead on the land cover dataset of over 10 billion pixels.

We also attempted directly comparing output label frequencies ($\mu_{\ell}$) to the NLCD class means $\eta_{\ell,z}$ using $L^2$ loss, as well as using an $L^1$ criterion (Lempitsky & Zisserman (2010)). In each case, the model converged to one that either predicted near-uniform class distributions at every pixel or always predicted the same class, giving accuracies below 30%. Interestingly, this occurred even when training was initialized with a well-performing trained model. These results indicate that the log-variance term in our criterion (7) is essential. (In these experiments, we used the same sample reweighting as in our super-res training and did a search through learning rates within a factor of 1000 of our baseline model learning rate.) Other approaches are discussed in Appendix C.

**Results.** The results for the baseline weakly supervised models and our SR models are shown in the first half of Table 1. We separately report overall results and results in NLCD blocks labeled with "Developed" (urban) low-resolution classes, which are the main source of errors for all models. Second, the Jaccard score is a more telling measure of classification quality than overall accuracy, which is dominated by large, easy-to-classify homogeneous areas (*e.g.*, forests) and gives little weight to accuracy on impervious surfaces. Thus the most important single metric is Jaccard score in developed classes (in italics in Table 1). In these areas, our SR-only model tends to outperform the baselines (see *second goal* below).

*First goal:* In the second half of the table, the HR only model serves as an upper bound for what is achievable by models that use only low-res data. Unsurprisingly, models trained only on low-

| | Maryland 2013 test region | | | | Chesapeake 2013 test region | | | | Chesapeake 2014 test region | | | |
|---|---|---|---|---|---|---|---|---|---|---|---|---|
| | all | | developed | | all | | developed | | all | | developed | |
| | acc% | iou% | acc% | iou% | acc% | iou% | acc% | iou% | acc% | iou% | acc% | iou% |
| Models trained on test geographical regions, without using high-resolution labels | | | | | | | | | | | | |
| Hard naïve | 83.5 | 70.1 | 58.2 | *38.5* | 87.7 | 68.0 | 63.4 | *40.2* | 88.1 | **63.6** | 68.2 | *46.4* |
| Soft naïve | **85.5** | 71.4 | 65.1 | *45.6* | **87.9** | 66.7 | 65.6 | *42.7* | **88.6** | 62.9 | 70.2 | ***48.3*** |
| EM | 73.9 | 40.0 | 59.9 | *32.2* | 82.3 | 42.1 | 60.0 | *32.0* | 81.7 | 40.4 | 61.9 | *33.0* |
| *SR* | 82.6 | **71.7** | 74.3 | *49.7* | 87.0 | **68.2** | 73.4 | *47.4* | 82.0 | 57.4 | 73.4 | *48.2* |
| Models using high-resolution labels in Maryland 2013 training set (more than $10^{10}$ labeled pixels) | | | | | | | | | | | | |
| HR only | **91.1** | **82.4** | 80.7 | ***64.9*** | 87.9 | 71.8 | 71.8 | *54.4* | 67.2 | 40.4 | 71.7 | *48.7* |
| *HR + SR* | 90.8 | 81.9 | 79.9 | *63.4* | **89.0** | **73.3** | 78.2 | *55.5* | **82.4** | **56.7** | 77.2 | *57.5* |

Table 1: Accuracies and Jaccard, or average intersection over union (IOU), scores on several data sets and models. Note that *we train models with high-resolution data from only the Maryland 2013 training region and low-resolution data from the region on which they are tested*. We give both overall metrics and those on areas labeled with NLCD "Developed, {Open, Low, Medium, High} Intensity" classes.

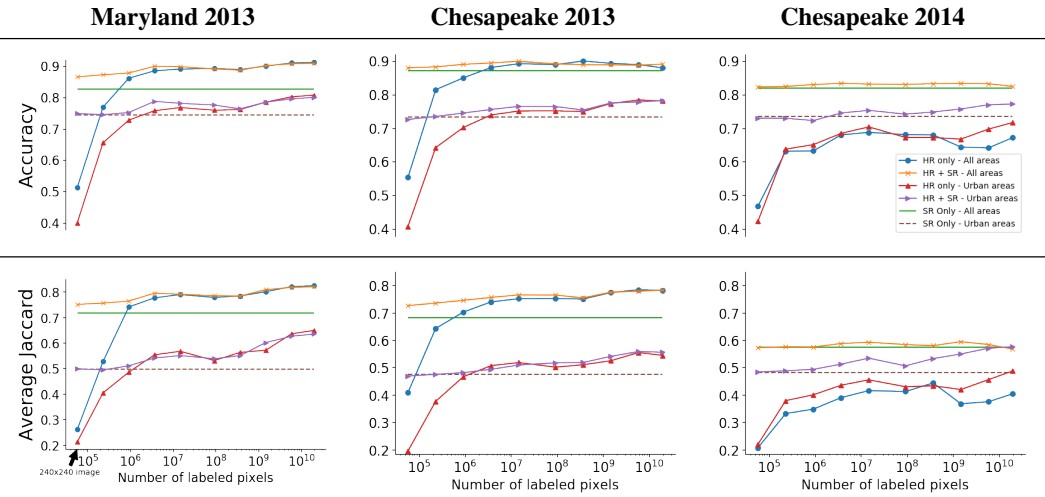

Figure 5: The effect of adding high-resolution data in training of super-resolution models. We show the baseline (HR only and SR only) and the results of HR+SR models with varying number of high-res label data seen in training, both overall and in developed classes (as in Table 1). All results presented are average of 5 experiments with different random samples of high-res data.

res data only are less accurate than those trained on high-res on the same region (Maryland 2013). Using low-res data together with high-res data adds uncertainty in training and slightly worsens results in Maryland, where high-res training data was used. However, adding low-res data from the test area allows our model to adapt to new geographies, with performance in developed areas in the two Chesapeake sets comparable to that in the original (Maryland) set. Furthermore, the SR-only model, not given high-res guidance, often produces segmentation that better match the true color segments and fine features of the images – see Fig. 4, an example from Maryland 2013).

*Second goal:* Overall, our super-res model performs better than the weakly supervised baselines. The "naïve" training criteria and the EM method perform especially poorly in developed classes. Indeed, while in classes such as "Open Water" or "Deciduous Forest", most pixels are labeled with the majority class label, in developed areas, the mean distributions are rather flat – "forest", "field", and "impervious" occur with nearly equal frequency in the "Developed, Low Intensity" class (see Table 4 in the appendix). Thus a model would prefer to make a highly uncertain prediction at every pixel in such a patch, rather than classify each pixel with confidence.

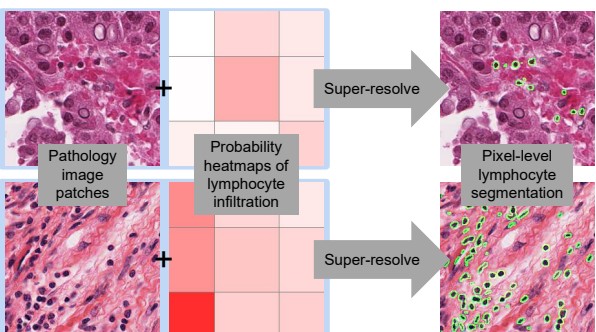

Figure 6: Our method is able to super-resolve the low-resolution probabilities of lymphocyte infiltration into pixel-level lymphocyte segmentation. Lymphocytes are dark, rounded small cells. Our method gives reasonable lymphocyte segmentation results (in green contours).

*Third goal:* Figure 5 shows how results of a model trained with our super-resolution technique improve when high-resolution labels are added in training. Performance gradually increases over the super-res baseline as more high-resolution data is seen. Models trained on both high-resolution and low-resolution data outperform HR-only models in all test sets *even when a small number ($10^6$ pixels $= 1\,\mathrm{km}^2$) of high-resolution pixels is seen in training*. In the Chesapeake 2014 dataset, HR+SR continues to far outperform HR-only even when all high-res data is used, and the metrics of HR+SR in developed classes even exceed those of HR-only overall. This demonstrates that super-resolution models can readily be used to create fine-scale labels in new geographies with only a small amount of strong supervision.

The full HR+SR accuracy of $89\%$ on the Chesapeake 2013 dataset is in fact close to the estimated accuracy (90%) of the "ground truth" labels over the entire Chesapeake region (Chesapeake Bay Conservancy (2017)) based on the same aerial imagery, which were themselves produced by a much more labor-intensive semiautomatic process (Chesapeake Bay Conservancy (2016)).

## 3.2 LYMPHOCYTE SEGMENTATION

We apply our method for lymphocyte segmentation in pathology images. Lymphocytes are a type of white blood cell that play an important role in human immune systems. Quantitative characterization of tumor infiltrating lymphocytes (TILs) is of rapidly increasing importance in precision medicine (Barnes et al. (2018); Finn (2008); Thorsson et al. (2018)). With the growth of cancer immunotherapy, these characterizations are likely to be of increasing clinical significance, as understanding each patient's immune response becomes more important. However, due to the heterogeneity of pathology images, the existing state-of-the-art approach only classifies relatively large tumor regions as lymphocyte-infiltrated or not. We show that our method is able to super-resolve the low-resolution probabilities of lymphocyte infiltration, given by the existing method (Saltz et al. (2018)), into pixel-level lymphocyte segmentation results. We illustrate this application in Figure 6.

**Datasets and training.** A typical resolution of pathology whole slide images is 50k$\times$50k pixels with 0.5 microns per pixel. An existing method (Saltz et al. (2018)) generated a probability heatmap for each of the 5000 studied whole slide images: every $100\times100$ pixel region was assigned a probability of being lymphocyte infiltrated. We use these probability heatmaps as low-resolution ground truth labels and super-resolve them into high-resolution (pixel-level) lymphocyte segmentation. To evaluate the segmentation performance, we use the lymphocyte classification dataset introduced in Hou et al. (2018). This dataset contains 1786 image patches. Each patch has a label indicating if the cell in the center of the image is a lymphocyte or not.

**Baseline models.** In addition to the Hard naïve and Soft naïve methods, we compare with the published models (Hou et al. (2018)) which are trained for lymphocyte classification in a supervised fashion. In particular:

- HR SVM: The authors first segment the object in the center of pathology patch with a level-set based method (Zhou et al. (2017)). Then they extract hand-crafted features such as the area, average color, roundness of the object (Zhou et al. (2017)). Finally they train an SVM (Chang & Lin (2011)) using these features.

- HR: Hou et al. (2018) directly train a CNN to classify each object in the center of image patches. This can be viewed as a CNN trained using high-resolution labels, although only the label of the center pixel is given.

- HR semi-supervised: Hou et al. (2018) initialize a HR CNN using a trained sparse convolutional autoencoder. Then the authors train the CNN to classify each object in the center of image patches.

Because all baseline CNNs require supervised data, they are all evaluated using four-fold cross-validation on the aforementioned dataset of 1786 image patches.

**Label super-resolution.** To use the low-resolution probability map as labels, we quantize the probability values into 10 classes with ranges $[0.0, 0.1), [0.1, 0.2), \ldots, [0.9, 1.0]$. In each low-resolution class, we sampled 5 regions labeled with this class and visually assessed the average number of lymphocytes, based on which we set the expected ratio of lymphocyte pixels in a given region ranging from 0% to 40%. With this joint distribution between low-resolution and high-resolution labels, we train our super-resolution network on 150 slides with low-resolution labels randomly selected from the 5000 slides. To guide our algorithm to focus on lymphocyte and non-lymphocyte cells instead of tissue background, we assign labels to 20% of the pixels in each input patch as non-lymphocyte, based on color only – pathology images are typically stained with Hematoxylin and Eosin, which act differently on nuclei of cells and cytoplasm (background), resulting in different colors. In terms of the network architecture, we apply the same U-Net as in the land cover experiment.

**Results.** We present quantitative results, obtained using the lymphocyte classification dataset, in Table 2. A testing image patch is classified as lymphocyte/non-lymphocyte by our method if its center pixel is segmented as lymphocyte/non-lymphocyte respectively. Our method performs as well as the best-performing baseline method with cell-level supervision.

|  | HR SVM | HR | HR semi-supervised | Hard naïve | Soft naïve | *SR* |
|---|---|---|---|---|---|---|
| AUC | 0.7132 | 0.4936 | **0.7856** | $0.5000^{\dagger}$ | 0.6254 | **0.7833** |

Table 2: Area Under receiver operating characteristic Curve (AUC) results of super-resolving low-resolution lymphocyte infiltration probability maps to individual lymphocyte segmentation, on the lymphocyte classification dataset from Hou et al. (2018). A testing image patch is classified as lymphocyte/non-lymphocyte by our method if its center pixel is segmented as lymphocyte/non-lymphocyte respectively. All HR baseline methods are directly evaluated on the classification dataset by four-fold cross-validation and reported by Hou et al. (2018). Our weakly supervised method performs effectively as well as the best-performing baseline method with cell-level supervision. $^{\dagger}$: Hard naïve achieves 0.50 AUC because there is no positive HR label, due to hard label assignment.

## 4 CONCLUSIONS

We proposed a label super-resolution network which is capable of deriving high-resolution labels, given low-resolution labels that do not necessarily match the targeting high-resolution labels in a one-to-one manner – we only assume that the joint distribution between the low-resolution and high-resolution classes is known. In particular, we train a network to predict high-resolution labels, minimizing the distance/divergence between two distributions: distribution of predicted high-resolution labels and expected distribution suggested by the low-resolution labels. We applied our method in two real-world applications where high res labels are very expensive to obtain compared to low res labels, and achieved similar or better results compared to the conventional fully supervised methods trained on high-resolution labels. We also show how combining low and high res labels leads to better generalization to out-of-sample test sets.

Although the main assumption of the model is that the joint distribution over coarse and fine labels is known, the model is in fact robust to errors in estimates of these distributions, as we discuss in Appendix F. There we show that these joint distributions can be acquired or inferred in a variety of ways, thus making label super-resolution widely applicable, including beyond computer vision.

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

## APPENDICES

## A MODEL ARCHITECTURE AND TRAINING DETAILS

The model cores and label counting modules were implemented in the CNTK (Seide & Agarwal (2016)) framework and trained using RMSProp (Hinton et al.) with an initial learning rate of $10^{-3}$ decaying to $10^{-6}$ by a factor of 10 per 6000 minibatches. Each minibatch contained 20 patches of $240 \times 240$ pixels each. Patches were sampled inversely to the frequency of occurrence of their NLCD (low-resolution) classes. The training loss converges after 14k minibatch iterations. As the network architecture, we use a U-Net (Ronneberger et al. (2015)) with 4 down-sampling and 4 up-sampling layers. After each down-sample or up-sample, we apply three convolutional layers with 32 to 64 filters. We apply batch normalization (Ioffe & Szegedy (2015)) before every activation function, except for the final (pre-softmax) output.

Results from some iterations of training appear in Figure 7.

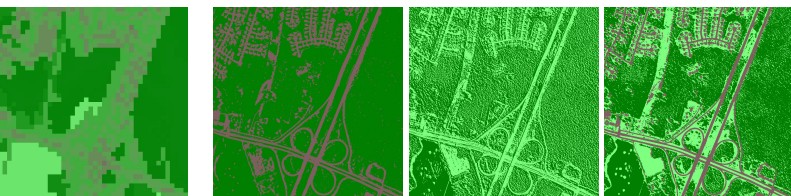

Figure 7: Left: the assignment to all pixels of the distribution over labels that corresponds to the NLCD patches containing them. Next: the inferred land cover map $y$ for an location near Richmond, Virginia, at 1m resolution at three different stages of training. As the learning progresses (on the entire state of Maryland), the labels get refined. Interestingly, the model flips the interpretations of the land use multiple times until it converges to the approximately correct mix of labels. Our model allows for this as the constraints on assignments for each input image are soft, as opposed to the models that enforce the constraints in each iteration of learning, *e.g.*, (Papandreou et al. (2015)).

## B NLCD DATA DESCRIPTION

The National Land Cover Database (Homer et al. (2015)), is the result of a joint effort of several US Governmental Agencies, and provides low-resolution (30 meter) land cover data covering the entire United States every 5 years. In this dataset land cover is represented by 16 classes, 15 of which are present in the Chesapeake Bay watershed for which we also have high-resolution land cover data (Chesapeake Bay Conservancy (2016)). Table 3 gives a description of each of the 15 NLCD classes we consider. Our proposed label super-resolution method relies on a joint distribution between the coarse NLCD classes, and the high-resolution land cover classes. As we show in Section 2, this type of distribution can be estimated from the descriptions of each NLCD class, but it can also be derived based on ground truth high resolution data. Table 4 shows this joint distribution calculated from the high-resolution data from the state of Maryland.

**Joint distribution between NLCD classes and 4 high-resolution land cover classes.** In Table 4, we show that for each low-resolution region ($30 \times 30$ pixels) with a given class (open water, etc.), the expected percentage ($\mu$) of each land cover class (water, forest, field, impervious), and the related standard deviation ($\sigma$). For example, regions with NLCD class "Developed, Open Space" in average have 42% of the pixels that are forest, and 46% of the pixels that are field.

## C ADDITIONAL EXPERIMENTS ON LAND COVER SEGMENTATION

The $(\rho, \sigma)$ in Table 4 can be viewed as *interval* constraints, similar to the ones used in (Pathak et al. (2015)) and the EM-adapt version of (Papandreou et al. (2015)). That is, we expect the frequency of high-res label $\ell$ in a block of low-res label $z$ to fall in an interval $(\eta_{\ell,z} - \rho_{\ell,z}, \eta_{\ell,z} + \rho_{\ell,z})$, where the $\eta$ and $\rho$ are either derived from aligned (high-res, low-res) data, as in our experiments or manually

| NLCD Class | Description |
|---|---|
| **Open Water** | open water, generally with less than 25% cover of vegetation or soil. |
| **Developed, Open Space** | mixture of some constructed materials, but mostly vegetation in the form of lawn grasses. Impervious surfaces account for less than 20% of total cover. These areas most commonly include large-lot single-family housing units, parks, golf courses, and vegetation planted in developed settings for recreation, erosion control, or aesthetic purposes. |
| **Developed, Low Intensity** | mixture of constructed materials and vegetation. Impervious surfaces account for 20% to 49% percent of total cover. These areas most commonly include single-family housing units. |
| **Developed, Medium Intensity** | mixture of constructed materials and vegetation. Impervious surfaces account for 50% to 79% of the total cover. These areas most commonly include single-family housing units. |
| **Developed, High Intensity** | highly developed areas where people reside or work in high numbers. Examples include apartment complexes, row houses and commercial/industrial. Impervious surfaces account for 80% to 100% of the total cover. |
| **Barren Land** | bedrock, desert pavement, scarps, talus, slides, volcanic material, glacial debris, sand dunes, strip mines, gravel pits and other accumulations of earthen material. Generally, vegetation accounts for less than 15% of total cover. |
| **Deciduous Forest** | dominated by trees generally greater than 5 meters tall, and greater than 20% of total vegetation cover. More than 75% of the tree species shed foliage simultaneously in response to seasonal change. |
| **Evergreen Forest** | dominated by trees generally greater than 5 meters tall, and greater than 20% of total vegetation cover. More than 75% of the tree species maintain their leaves all year. Canopy is never without green foliage. |
| **Mixed Forest** | dominated by trees generally greater than 5 meters tall, and greater than 20% of total vegetation cover. Neither deciduous nor evergreen species are greater than 75% of total tree cover. |
| **Shrub/Scrub** | dominated by shrubs; less than 5 meters tall with shrub canopy typically greater than 20% of total vegetation. This class includes true shrubs, young trees in an early successional stage or trees stunted from environmental conditions. |
| **Grassland/Herbaceous** | dominated by gramanoid or herbaceous vegetation, generally greater than 80% of total vegetation. These areas are not subject to intensive management such as tilling, but can be utilized for grazing. |
| **Pasture/Hay** | grasses, legumes, or grass-legume mixtures planted for livestock grazing or the production of seed or hay crops, typically on a perennial cycle. Pasture/hay vegetation accounts for greater than 20% of total vegetation. |
| **Cultivated Crops** | used for the production of annual crops, such as corn, soybeans, vegetables, tobacco, and cotton, and also perennial woody crops such as orchards and vineyards. Crop vegetation accounts for greater than 20% of total vegetation. This class also includes all land being actively tilled. |
| **Woody Wetlands** | forest or shrubland vegetation accounts for greater than 20% of vegetative cover and the soil or substrate is periodically saturated with or covered with water. |
| **Emergent Herbaceous Wetlands** | perennial herbaceous vegetation accounts for greater than 80% of vegetative cover and the soil or substrate is periodically saturated with or covered with water. |

Table 3: Descriptions of NLCD classes occurring in the Chesapeake Bay region, from Homer et al. (2015).

| NLCD class | freq. | water η | water ρ | forest η | forest ρ | field η | field ρ | imperv. η | imperv. ρ |
|---|---|---|---|---|---|---|---|---|---|
| Open water | 17.8% | .97 | .15 | .01 | .06 | .01 | .06 | .02 | .13 |
| Developed, Open Space | 7.1% | .00 | .05 | .42 | .34 | .46 | .33 | .11 | .13 |
| Developed, Low Intensity | 3.1% | .01 | .06 | .31 | .24 | .34 | .21 | .35 | .18 |
| Developed, Medium Intensity | 1.5% | .01 | .07 | .14 | .17 | .21 | .19 | .63 | .22 |
| Developed, High Intensity | .7% | .01 | .07 | .03 | .07 | .07 | .14 | .89 | .17 |
| Barren Land (Rock/Sand/Clay) | .4% | .09 | .26 | .13 | .26 | .45 | .41 | .32 | .40 |
| Deciduous Forest | 26.2% | .00 | .03 | .92 | .19 | .06 | .16 | .01 | .07 |
| Evergreen Forest | 2.3% | .00 | .03 | .94 | .18 | .05 | .16 | .01 | .05 |
| Mixed Forest | 1.3% | .01 | .05 | .92 | .18 | .06 | .15 | .02 | .06 |
| Shrub/Scrub | 1.1% | .00 | .05 | .71 | .35 | .26 | .33 | .03 | .09 |
| Grassland/Herbaceous | .3% | .01 | .09 | .38 | .40 | .54 | .39 | .07 | .18 |
| Pasture/Hay | 1.7% | .00 | .02 | .11 | .21 | .86 | .23 | .03 | .09 |
| Cultivated Crops | 16.9% | .00 | .03 | .11 | .22 | .86 | .24 | .03 | .09 |
| Woody Wetlands | 7.8% | .01 | .07 | .90 | .22 | .08 | .21 | .00 | .03 |
| Emergent Herbaceous Wetlands | 2.7% | .11 | .21 | .07 | .22 | .81 | .29 | .01 | .05 |

Table 4: Means and standard deviations of high-resolution land cover class frequencies within blocks labeled with each NLCD class, computed on the state of Maryland.

set. One option, which we take in our experiments below, is to manually extract these parameters from the NLCD class specifications.

However, direct use of interval constraints in the statistics matching module of our network fails to produce the label super-resolution results described in the main text. While the bounds may satisfied, the true distributions of high-res classes given NLCD labels is obscured. For example, the "Open water" class denotes anywhere between $75\%$ and $100\%$ of water in the $30 \times 30$ block (see Table 3). But all values in this range are not equally likely, as Table 2 shows: In fact, on average $97\%$ of the $1m \times 1m$ pixels are water. This is because the majority of the blocks labeled "Open water" are in the middle of a lake, river, or ocean, and only a small fraction has a smaller percentage of water. Thus it is undesirable to enforce them strictly in each image as (Papandreou et al. (2015))'s EM-Adapt. Similar observations hold for most classes – both those with a single frequent high-res label (Open Water, Evergeen Forest) and not (Developed) – so a Gaussian model fits the data better.[4]

The other problem with direct use of models in (Pathak et al. (2015); Papandreou et al. (2015)) is that all NLCD classes $c$ overlap with others in one or more 1m land use labels $y$, especially in case of urban classes, and the differences are slight. In particular, learning of the (fine) impervious surface label is based on very slight variation in the four (coarse) developed classes, which often contain more field and tree pixels than impervious pixels.

In Table 5 we show our Gaussian models for statistics matching lead to dramatically better label super-resolution results than direct use of interval constraints, where overall loss is represented as minimum $L^2$ distance from the output distributions to the target interval (and no penalty is given when label counts fall within the target interval).

# D    LAND COVER CHANGE DETECTION

Given two signals (raw images or land cover predictions) in the same geographical location obtained from two different years, we scan through $240 \times 240$-meter windows and compare the two signals in the following way to detect land cover change:

1. To reduce color shift in each channel, we calibrate signals from one year to another. In particular, we apply a least square linear regression model, taking each pixel from the source

---

[4]In fact, a multi-modal distribution might be even more appropriate, and could be accommodated in our approach that separates the label counting and statistics matching modules.

|  | all acc% | all iou% | developed acc% | developed iou% |
|---|---|---|---|---|
| HR only | 90.5 | 78.1 | 78.1 | 62.2 |
| *SR only* | 80.5 | 64.3 | 71.4 | 51.2 |
| *interval* | 72.7 | 48.1 | 54.2 | 32.9 |

Table 5: Comparison of super-resolution results using our approach and modifying our approach to use interval constraints, trained on evaluated on a variant of the Maryland 2013 dataset.

     signal as input (multi-channeled, ignoring spatial information), to predict the corresponding pixel value in the target signal. Then we use the predicted pixel values as calibrated signals.

2. After calibration, we compute the mean absolution difference across all pixels, between two signals. If the resulting value is above a certain threshold, we report this detected change.

Figure 3 shows detected land cover change. We conclude that using the raw NAIP satellite images as signals yields poor results, whereas using land cover predictions by our model as signals yields reasonable land cover change detection results.

## E    LAND COVER WEB APPLICATION

We created a web application - shown in Figure 8 and accessible online at `http:// landcovermap.eastus.cloudapp.azure.com:4040/` - that lets users interactively query our best models by clicking in a map interface. This tool lets us easily test the qualitative performance of our models in a variety of settings (*e.g.*, in urban vs. rural areas), compare the outputs of different models side-by-side, and communicate our results with domain experts that will ultimately use this landcover data. We summarize the functionality of the tool as follows:

1. Users can "paint" land cover predictions onto an ESRI World Imagery basemap and adjust the opacity of the predictions to see how the predictions match up with the underlying imagery (note that the predictions are not made with the ESRI World Imagery but with NAIP images shown in the right sidebar).

2. After selecting an area for prediction on the map, users can switch between which model output is displayed by clicking on the different predictions under the "Land Cover Predictions" heading in the right sidebar. Our tool currently shows predictions from our best performing **HR+SR** model described in Section 3.1 on the left, and predictions from a model trained with US wide data not described in this paper on the right.

3. Users can manually set a weighting scheme used to modify the model's predictions by adjusting the sliders shown in the sidebar. These weights will be applied to all predictions (clicks) after they are set.

## F    ON OTHER APPLICATIONS:
## FOREGROUND-BACKGROUND SEGMENTATION IN A BOUNDING BOX

It may seem that having available a joint distribution over coarse and fine labels is a rare situation. However, we can put ourselves into that situation in multiple ways, because the learning is often robust to errors in estimates of these distributions. In neither one of our two main applications was the estimate of the distribution exactly right, and the results were fairly robust to variation in the distribution models.

For example, for land cover classification, we could set the target distributions based on the descriptions in the NLCD specification (Table 3). Indeed, we found that this gave similar results, although more noise was seen in classes like "Water" and Evergreen Forest" where the specification allows for a wide interval (*e.g.*, [0.75,1], translated into $\mu = 0.875$ and $\sigma = 0.25/\sqrt{(12)}$) but the true

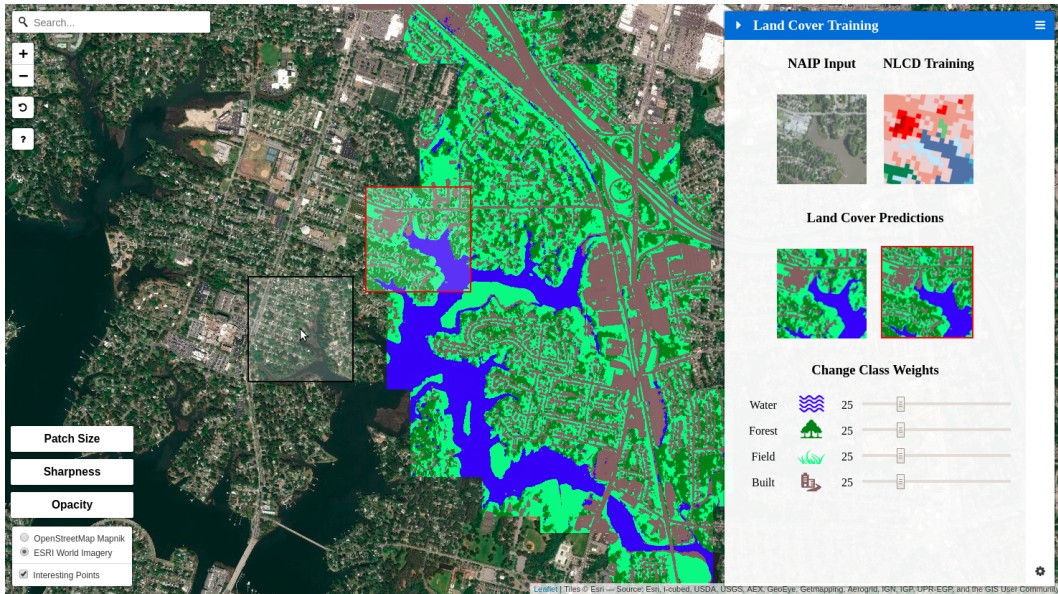

Figure 8: Web application that lets you interactively query our best **HR+SR** model for any area in the United States. The application shows the NAIP input imagery which the model is run on, the corresponding low-resolution NLCD labels for that area, and our high-resolution model output. Users can click on the map which will run model inference on-the-fly and "paint" the resulting land cover predictions over the ESRI basemap. This tool can be found online at `http://landcovermap.eastus.cloudapp.azure.com:4040/`.

mean is much closer to 1 (cf. Table 4). Furthermore, this distribution can be tuned by hand (forcing the "Water" class to have higher proportion of water than what was in the NLCD description, for example). If there are only a handful of coarse and fine-grained labels, then such experimentation is not unreasonable. This means that many application scenarios can benefit from our approach.

As a way of an example, consider segmentation of cropped pedestrians. We show here that by providing our algorithm with a very rough block guidance, we can get results close to what we can achieve using fine-grained hand segmentation as training labels. For example, in one experiment below, the labels in each block of a $6 \times 10$ grid over the image is simply assumed to follow one of three uniform distributions (more than $\frac{2}{3}$ background, more than $\frac{2}{3}$ foreground, and between $\frac{1}{3}$ and $\frac{2}{3}$ foreground). Such labeling is less expensive for crowd-sourcing as it leads to faster and more consistent labeling across workers.

This problem then falls squarely into weakly supervised segmentation methods widely studied in computer vision. These methods excel when there is some spatial structure of the task that can be exploited. Our problem formulation can capture these cases, although in the main text we focus on the cases where traditional weakly supervised models are ill-suited. We briefly demonstrate this by performing foreground-background segmentation of pedestrians taken from the Cityscapes and CityPersons datasets (Cordts et al. (2016); Zhang et al. (2017)). We extract images from the bounding boxes in the "pedestrian" class, and use 5700 images of a standardized size of $82 \times 200$ in training.

As in the rest of the experiments, we first use a spatially invariant small U-Net model with 4 up-sampling and down-sampling layers and 16 filters in each layer. In training, we divide each $82 \times 100$ image into eight $41 \times 50$ blocks and give the counts of foreground pixels in each block as input to the statistics matching layer. For the coarse target distributions, we use the true label frequencies as means, with a fixed variance ($\sigma^2 = 0.1$). (Alternatively, we could quantize the frequencies into buckets $[0.1, 0.2)$, $[0.2, 0.3)$, etc., as in the lymphocyte application – see Section 3.2 – but we found that this does not affect the results.)

We train three models: using our super-resolution technique, using $L^2$ distance between frequencies as the loss, and using an interval constraint with a radius of 0.2 (see Appendix C). We compare the results with baseline models trained on high-resolution data with pixelwise cross-entropy and $L^2$ losses.

Because our main applications' focus on *spatially invariant* segmentation of large images, we again use the core U-Net architecture, which applies the same convolutional filters uniformly across images. The core segmentation network could simply be replaced with any other model. In particular, for bounding box segmentation, a better suited core network would have different weights going to different pixel labels at the output because spatial invariance is violated in this application: Pixels close to the edges, and especially corners of the box, are more likely to be background pixels.

The results are shown in the first five rows of Table 6; some example segmentations can be seen in Figure 9.

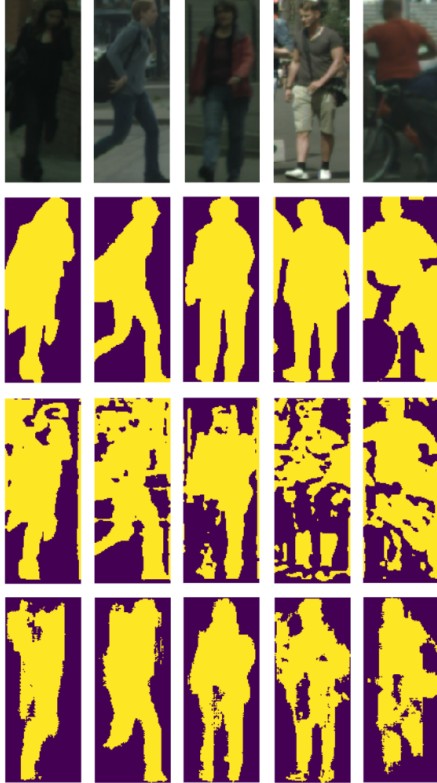

Figure 9: Segmentations of pedestrians from the Cityscapes dataset, small U-Net model. Rows top to bottom: input image, ground-truth segmentation, super-res model output, high-res model output.

In an extension to these experiments, we used a larger U-Net variant (24 filters in each layer and a fully connected bottleneck layer) to train pedestrian segmentation models given quantized finer data. Each training image was divided into a $6 \times 10$ grid of $13 \times 20$ blocks, each of which was reduced to one of three labels depending on the frequency of foreground (pedestrian pixels): $[0, \frac{1}{3})$, $[\frac{1}{3}, \frac{2}{3})$, $[\frac{2}{3}, 1]$ – roughly, "background", "boundary", and "foreground". The super-resolution model was trained as above, yielding the example segmentations in Fig. 10. The results are in the last two rows of Table 6. Although the super-res model sees no low-resolution data in training and thus appears to be more sensitive to fine features, it locates object boundaries comparably to the high-resolution model.

While pedestrian segmentation is no longer considered too challenging a problem due to the size of the existing labeled data, this example illustrates that our technique can be used to reduce the costs of crowdsourcing to acquire such labels for new object classes. Furthermore, it is possible to ac-

| core | criterion | training label dim. | acc% | iou% |
|------|-----------|---------------------|------|------|
| small | *super-res* | $2 \times 4$ | **68.5** | **49.4** |
| small | interval | $2 \times 4$ | 67.4 | 46.1 |
| small | $L^2$ distance | $2 \times 4$ | 66.4 | 48.4 |
| small | high-res ($L^2$) | $82 \times 200$ | 75.8 | 58.0 |
| small | high-res (cross-ent) | $82 \times 200$ | 74.4 | 59.2 |
| large | super-res | $6 \times 10$ | 76.3 | 59.8 |
| large | high-res ($L^2$) | $82 \times 200$ | 80.3 | 65.2 |

Table 6: Results of Cityscapes segmentation models.

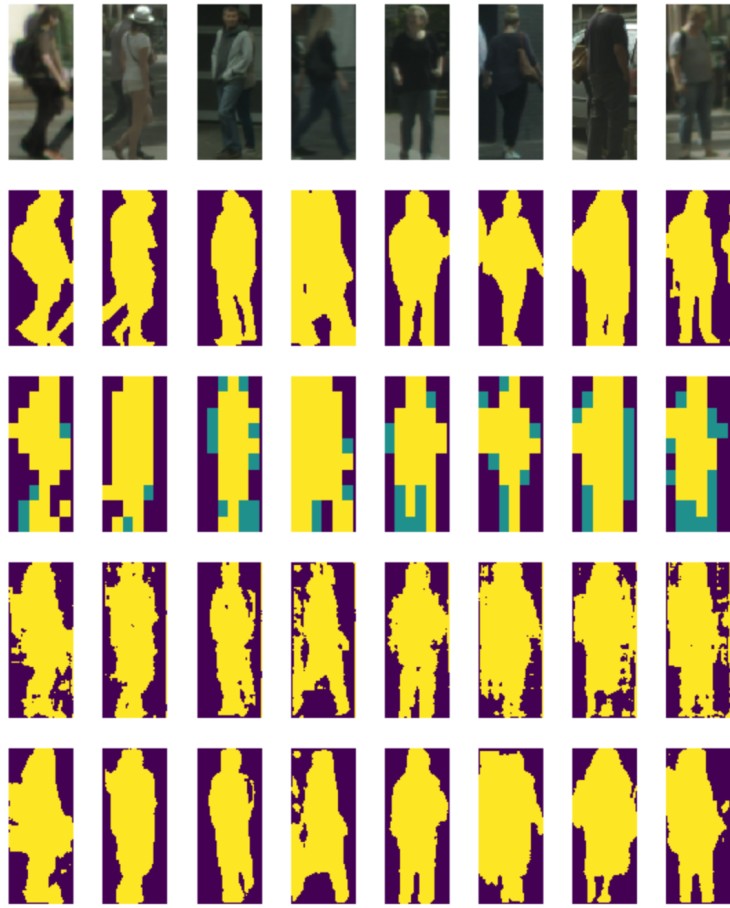

Figure 10: Segmentations of pedestrians from the Cityscapes dataset, large U-Net model. Rows top to bottom: input image, ground-truth segmentation, quantized frequencies of "pedestrian" pixels in $13 \times 20$ blocks (label data seen in super-res training), super-res model output, high-res model output.

quire such labeling automatically without crowdsourcing. For example, one of the co-segmentation techniques or class activation mapping techniques (Zhou et al. (2016)) can be used to gain an approximate segmentation from which these coarse labels can be derived.

Possibly the very first and the simplest unsupervised co-segmentation technique is the probabilistic index map model (PIM) of Jojic & Caspi (2004), later extended in various ways, *e.g.*, (Winn & Jojic (2005); Jojic et al. (2009)). The basic probabilistic index map model can be implemented in less than a dozen lines of code. It analyzes a collection of approximately registered images (such as object crops) using simple Gaussian color models. The model assumes that while the prior over class correspondences for image pixels is shared across the collection, the color palette (the means and variances of the Gaussian components only apply to a single image, so that each

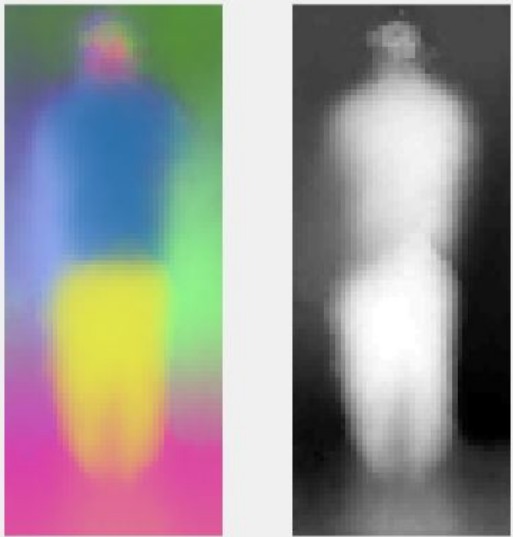

Figure 11: Probabilistic index map with 8 indices (segments) learned using (Jojic & Caspi (2004)) is shown on the left. The map is a prior for segmentation of individual images, each following a different palette: a different image-specific Gaussian mixture with 8 components. On the right, we show inferred grouping of the 8 segments into two – foreground and background, based on the assumption that the left edge and the right edge are most likely background.

image has its own palette). This model then discovers groupings of pixels (such as the torso area of the pedestrian images) that are consistently grouped into segments of uniform color across the image collection, even if that color can vary drastically from one image to another. Once the images are jointly (over)segmented like this into a fixed number segments (in our case, 8, as shown in Fig. 11), we can assume that the left and right edge of the crops are more likely to be background than foreground and assign these segments to the background and the rest to foreground (Fig. 11). This foreground/background assignment is performed probabilistically, *i.e.*, based on the statistics of segment uses along the crop edges. In addition, the consistency of segmentation is controlled though clipping the variances of the Gaussian models from below, as this encourages collection-generalizing over image-specific segmentations. Some examples of image segmentations are shown in Fig. 12, where we also demonstrate that such automatic segmentation can be used to estimate the $13 \times 20$ block distributions in the $6 \times 10$ grid and use them as such, or sort them into the three categories as in Fig. 10. While the PIM model on its own does not produce excellent fine-grained segmentation of individual images, as it cannot reason about texture features, it can produce reasonable approximations to the block segmentation that would allow our label super-resolution network to yield segmentations as discussed above.

Another application idea is to simply use positive and negative examples of pedestrians cropped from the Cityscapes dataset and assume that in positive examples at least 60% of pixels are pedestrians, while in negative crops of urban scenes, that percentage is less than 5%.

The intent of this toy example is to demonstrate that the label count constraints are abundant in computer vision, and show that label super-resolution models can be fairly robust to approximations of these constraints.

In fact, count constraints are not uncommon in applications beyond computer vision. A recent natural language processing paper (Srivastava et al. (2018)) addresses the problem of learning the features of important emails through soft guidance on groups of emails (emails from this person are usually important; emails from this domain are rarely important, emails sent between 9-10am are important 60% of the time, etc.). Similar situations arise in public health, when different studies provide different statistical summaries for groups of patients and the causes are not separated (*e.g.*, prevalence of certain tumors in patients of certain age, association with certain lifestyle choices, etc.). Analysis of individual detailed medical records could lead to learning to make fine-grained

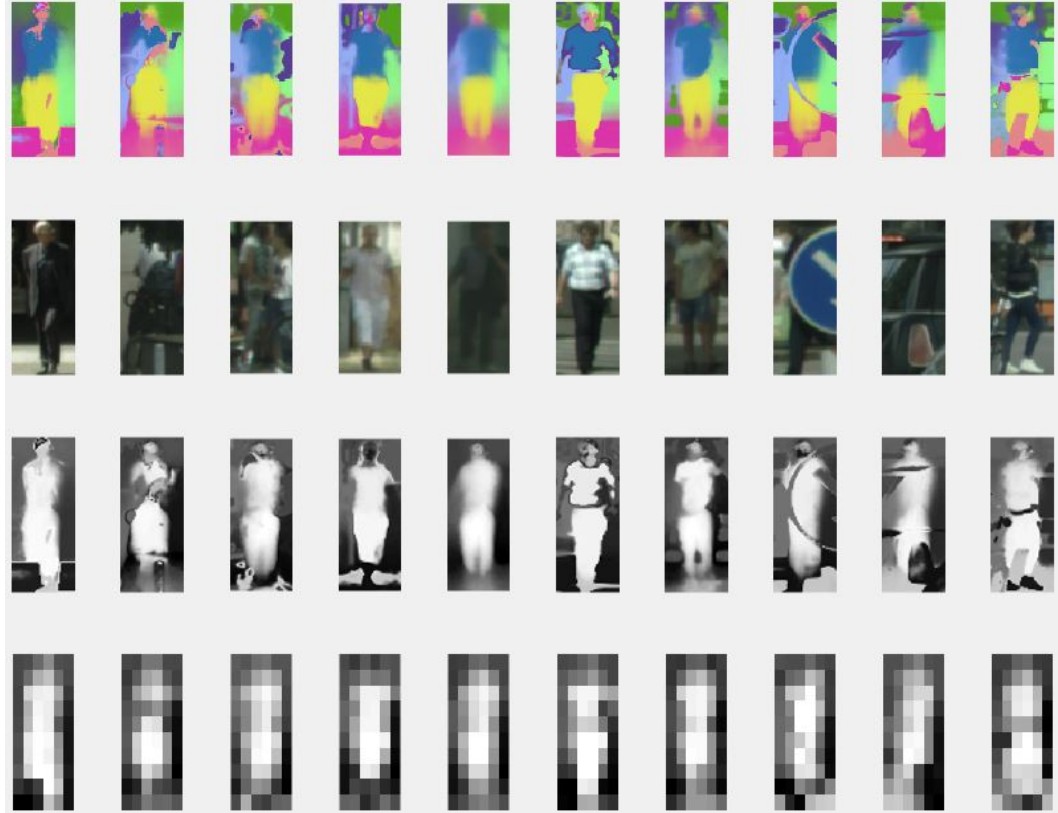

Figure 12: Unsupervised segmentation maps obtained using the probabilistic index map model from Fig. 11 are shown in the first row for images in the second row. The third row shows foreground/background segmentation based on the segment grouping in Fig. 11, and the final row shows block estimates of statistics on foreground pixel counts, which can then be used as coarse labels in Fig. 10. This is just one example in which coarse labels can be created in an unsupervised manner.

predictions for individual patients without having these labels in training. Our technique can be adapted to these applications as well.

