# OpenReview forum: "Label super-resolution networks"
_ICLR.cc/2019/Conference_

### Official Review · AnonReviewer2 · 2018-10-29
**Fun, useful, and well presented idea. Experimental results are convincing, too.**

**Rating:** 9
**Confidence:** 4

**Review:**

Paper summary:

This paper presents a deep-learning based method for super-resolving low-resolution labels into high-resolution labels given the joint distribution between those low- and high- resolution labels. This is useful for many semantic segmentation tasks where high-resolution ground truth data is hard and expensive to collect. Its main contribution is a novel loss function that allows to minimize the distance between the distribution determined by a set of model outputs and the corresponding distribution given by low-resolution label over the same set of outputs. The paper also thoroughly evaluates the proposed method for two main tasks, the first being a land cover mapping task and the second being a medical imaging problem.

For the land cover application, adding low-resolution data to high-resolution data worsens the results when evaluating on the geographic area from which the high-resolution data was taken. However, when testing the model on new geographic areas and only adding the low-resolution data from this new area in training makes significant improvements.

Generally the paper is very well written, well structured, all explanations are clear, examples and figures are presented when needed and convey helpful information for the reader. The overall idea is fun, original, useful (especially in remote sensing) and is presented in a a convincing way. All major claims are supported by experimental evaluation. There are nevertheless a few concerns:

Major Concerns:

On a conceptual level, the main concern is that the paper assumes we are given a joint distribution of low and high resolution labels, “where we are given the joint distribution P(Y,Z)”, which seems the main limitation of this method. In fact, to correctly estimat this joint distribution either requires additional knowledge about low-resolution data such as the example presented on the NCLD data : “For instance, the “Developed, Medium Intensity” class [...] of the coarse classes”, or it requires actual high-resolution labelled data to correctly estimate this joint distribution. I think the paper would greatly benefit from including a section that discusses the impact of this limitation.

Another point is footnote 3 on page 5. This argument is valid but it would be more convincing to give a thorough explanation on why the choice of the presented loss function is better compared to the KL divergence based loss function or at least some evidence that the two perform similarly when evaluating the method.

Minor Concerns:

-	“such as CRFs or iterative evaluation” I would include a citation on this type of work.
-	Format of some references in the text need to be corrected, e.g. “into different land cover classes Demir et al. (2018); Kuo et al. (2018); Davydow et al. (2018); Tian et al. (2018).”

---

> ### Author Response · Authors · 2018-11-15
> **Authors' response**
>
> [Modified Nov. 19 to reflect changes in the text.]
>
> Thank you for your thoughtful comments and questions. We have taken account of the Minor Concerns you raised.
>
> In response to the Major Concerns:
>
> We agree with your comment on P(Y,Z) and will incorporate discussion of both the generality and limitations of fixing a joint distribution. Please see the response to Reviewer3, part 1, above, as well as the updated Appendix F [formerly Appendix D], for discussion on this. In short, the estimated joint distribution need not be derived from high-resolution data: it could be specified a priori, tuned manually, derived from the output of another model, etc.
>
> Loss functions: We summarize the intuitive motivation in our response to Reviewer3, part 2, above.
>
> Qualitatively, the two loss functions have a similar form. They minimize:
> (first term) the L2 distance between observed and expected counts normalized by the expected (resp. expected+observed) variance in (7) (resp. KL);
> (second term) the variance at individual pixels.
> In (7), the added term in the denominator reduces the weight of the L2 distance when the model is uncertain (sigma is large). When the block size is large, the difference between the two functions becomes insignificant. However, when the block size is small, (7) punishes the model for predictions that are very certain but incorrect, so it must balance between high certainty (second term) and low certainty on unlikely predictions (first term).
>
> Quantitatively, if there are c classes, the maximum possible sigma2 occurs when all outputs are uniform over the classes and is equal to
> 1/Bk * 1/c * (1-1/c),
> In the land cover experiment, where Bk=900, and c=4, sigma2 is approximately 0.0002. If rho=0.03, as it may be for classes very unlikely to occur in given blocks (see Table 4), then rho2=0.0009, on the same order as sigma2. Thus we punish more for predictions that are certain but predict a class that is unlikely to occur. As predictions become more certain during training, sigma2 becomes insignificant.
>
> Indeed, in early experiments we found that beginning SR-only training with the KL distance sometimes led to pathological local minima, such as a single class always being predicted with high certainty. In contrast, it seems that using (7) in early training -- favoring uncertainty in unlikely predictions -- enables the behavior in Figure 7. However, if training is initialized with a well-performing model, the two criteria give similar results.

---

### Official Review · AnonReviewer1 · 2018-11-02
**Interesting approach, and unique use cases**

**Rating:** 6
**Confidence:** 4

**Review:**

This paper presents a method to super-resolve coarse low-res segmentation labels, if the joint distribution of low-res and high-res labels are known. The problem formulation and the proposed solution are valid, given the examples of land cover super-resolution and lymphocyte segmentation.
I like the paper in general, with the following concerns/thoughts:
1. While matching the divergence of low-res and high-res segmentations, will the model simply collapse and predict noisy boundaries? Or is it already the case, as can be seen in Figure 8 of Appendix? It seems possible that the model is learning high resolution noises. I suggest the authors to do more careful analysis on this.
2. I am curious to see if the proposed technique can be used in other aspects, like super-resolving the boundary of semantic segmentations.

---

> ### Author Response · Authors · 2018-11-15
> **Authors' response**
>
> [Modified Nov. 19 to reflect changes in the text.]
>
> Thank you for your thoughtful comments and questions.
>
> (1) In Figure 8 [now Figure 9; see also Figures 10-12], we see qualitatively that both the high-resolution and low-resolution models are sensitive to small-scale input features, and the low-resolution model indeed has little punishment for small-scale *label* errors when data is given at a scale of 8 numbers per image. Yet, our results demonstrate that a model that sees no high-resolution data can learn to (a) be sensitive to shape and (b) make highly certain predictions around boundaries (cf. Fig. 4).
>
> This also depends on the capacity of the core segmentation model. In principle, if it is highly expressive, it could learn to recognize the blocks and fill in the labels inside the block to fit the frequencies without regard to the input features. This did not happen in our experiments, partly because the neural networks are difficult to overtrain.
>
> (2) Please see the response to Reviewer3 above regarding the uses of our method beyond the setup of the land cover example, as she or he raised closely related questions. We have updated Appendix D [now Appendix F] with an example of super-resolving coarse segmentations and discussion of other approaches to obtaining coarse labels.

---

### Official Review · AnonReviewer3 · 2018-11-02
**A very well written paper with substantial and well organized experimental content, but overall a bit too narrow in scope and technical contribution**

**Rating:** 7
**Confidence:** 4

**Review:**

The authors present a technique to exploit low resolution labels from a space Z to provide weak supervision to a semantic segmentation network which predicts high resolution labels from a different space Y, assuming that a joint distribution of Z, Y is known a-priori.

The paper is very well written and easy to follow, the main contribution is clearly and rigorously explained in the technical section.
The technical contribution is somehow limited, but it is substantially validated by a very well organized and convincing experimental evaluation.
Overall, I have three main points of criticism (detailed in the following), which however aren't enough to not recommend this paper for acceptance.

Main cons:

1) At points, the paper reads more like a technical report about solving specific problems in land cover estimation and lymphocyte segmentation than a machine learning paper.
Many paragraphs are devoted to describe the specifics of these two problems and to design methods to overcome them.
The main technical contribution of the paper seems to be specifically tailored to solve the particular setup encountered in land cover estimation, i.e. two different sets of labels with different resolution on the same segmentation data, which ties to the next point.

2) One important limitation lies in the fact that the distribution p(c|z) needs to be known a-priori and somehow derived from additional problem-specific knowledge.
This is not an issue in the two tasks considered in the paper, but in my opinion it could severely limit the applicability of the proposed approach.
I think the paper would benefit from the inclusion of some discussion about how this limitation could be overcome.

3) It's not very clear to me why the gaussian approximation with the specific mean and variance values defined in eq.4 would be a good approximation for p_net(c_lk|X).
Could the authors expand on this?

---

> ### Author Response · Authors · 2018-11-15
> **Authors' response, part 1**
>
> [Modified Nov. 19 to reflect changes in the text.]
>
> Thank you for your thoughtful comments and questions.
>
> In response to (1) and (2):
>
> Certainly, knowing the distributions p(c|z) is a prerequisite to using our method. In the problem we are considering, where high-resolution and low-resolution classes may not match one-to-one, one must establish at least a weak correspondence between the two kinds of classes -- else, one does not know anything about the meaning of the target (high-res) classes.
>
> In our main example, land cover mapping, high-resolution data is expensive and difficult to collect, but plenty of low-resolution data exists. However, our method is more general, as there are different potential sources of this distribution:
>
> - Labels given in coarse blocks with a known distribution (as the NLCD data in our land cover example). In fact, these need not be derived from any high-resolution data. For example, we could set the target distributions based on the descriptions in the NLCD specification (Table 3). Indeed, we found that this gave similar results, although more noise was seen in classes like "Water" and Evergreen Forest" where the specification allows for a wide interval (e.g., [0.75,1], translated into mu=0.875 and sigma=0.25/sqrt(12)) but the true mean is much closer to 1 (Table 4).
> Furthermore, this distribution can be tuned by hand (forcing the "Water" class to have higher proportion of water than what was in the NLCD description, for example). If there are only a handful of coarse and fine-grained labels, then such experimentation is not unreasonable.
>
> - Quantized density estimates from another model (as the coarse predictor output in our lymphocyte example).
>
> - A coarse segmentation provided by another model. We mock this by blurring the ground truth distribution in the Cityscapes pedestrians example, but this may come from the output of a coarser segmentation model, a class activation map coming from a classification model, etc. (as Reviewer1 seems to be suggesting).
>
> We have added a small extension to the Cityscapes pedestrians example, showing how we can super-resolve coarse segmentations. We have updated Appendix D [now Appendix F] with these results and revised the text to emphasize the applicability of our approach to different kinds of problems.
>
> We think that more general approaches to overcoming this limitation would be an interesting subject for future work. Potential directions are: (a) beginning with only rough priors on the distributions, estimate them by iteratively updating them with the label counts currently being predicted in blocks of each low-resolution class during training; (b) combine this with an unsupervised segmentation method to infer high-resolution classes, knowing they are distributed similarly in blocks of any given low-resolution class. In other words, we could estimate the joint distribution with EM. However, in most applications some knowledge of the relationship between classes is available, and, as discussed above, even weak or hand-tuned priors p(c|z) are often sufficient (depending, of course, on the capacity of the core model and the ability of the gradient descent to vastly overtrain).
>
> We do think the method will be of interest to wide readership as there are many ways to adapt it to new applications. (We do see now that our focus on the two applications that most need this method may create impression that the idea is limited to these couple of applications, and we will address that in the writing.)

---

> ### Author Response · Authors · 2018-11-15
> **Authors' response, part 2**
>
> In response to (3):
>
> We view the output of the core network as a generative model of (hard) segmentations, where the label at a given pixel is drawn from the distribution given by the model’s output. A version of the central limit theorem implies that if one samples the label at each pixel, the count of pixels of a class c within a block, appropriately normalized, will follow an approximately Gaussian distribution whose mean and variance are the average mean and variance of the distributions at individual pixels (eq. 4).
>
> This point of view also leads naturally to the proposed loss function (eq. 7). Here we are maximizing the probability of the model producing the set of labels with the highest log-likelihood under both the network output and the known joint distribution of high-res and low-res labels. (In other words, we independently draw counts from p_net and p_coarse and choose the optimal counts conditioned on the two counts being equal.) On the other hand, the KL divergence mentioned in the footnote measures the expected log-likelihood under p_coarse of a sample count drawn from p_net.
>
> Please see the response to Reviewer2 below for more discussion of the statistics and loss functions.

---

> ### Comment · AnonReviewer3 · 2018-11-26
> **Reviewer response**
>
> The authors clarified the doubts I expressed in the review and properly answered all my questions.
> Given this, I confirm my positive rating.

---

### Meta-Review · Area_Chair1 · 2018-12-10

**Confidence:** 4
**Recommendation:** Accept (Poster)

**Metareview:**

This paper formulates a method for training deep networks to produce high-resolution semantic segmentation output using only low-resolution ground-truth labels. Reviewers agree that this is a useful contribution, but with the limitation that joint distribution between low- and high-resolution labels must be known. Experimental results are convincing. The technique introduced by the paper could be applicable to many semantic segmentation problems and is likely to be of general interest.